# Pending recovery in the strength of the meridional overturning circulation at 26°N

Ben. I. Moat[1], David. A. Smeed[1], Eleanor Frajka-Williams[1], Damien G. Desbruyères[2], Claudie Beaulieu[3], William E. Johns[4], Darren Rayner[1], Alejandra Sanchez-Franks[1], Molly O. Baringer[5], Denis Volkov[5,6], Laura C. Jackson[7], Harry L. Bryden[8]

1) National Oceanography Centre, University of Southampton Waterfront Campus, European Way, Southampton, SO14 3ZH, UK.

2) Ifremer, University of Brest, CNRS, IRD, Laboratoire d'Océanographie Physique et Spatiale, IUEM, Ifremer centre de Bretagne, Plouzané, 29280, France.

3) Ocean Sciences Department, University of California Santa Cruz, CA, USA.

4) Rosenstiel School of Marine and Atmospheric Science, University of Miami, Miami, FL, USA.

5) Atlantic Oceanographic and Meteorological Laboratory, NOAA, Miami, FL, USA.

6) Cooperative Institute for Marine and Atmospheric Studies, University of Miami, Miami, FL, USA,

7) Met Office, UK.

8) School of Ocean and Earth Science, University of Southampton Waterfront Campus, European Way, Southampton, SO14 3ZH, UK.

*Correspondence to*: Ben Moat (ben.moat@noc.ac.uk)

**Abstract.** The strength of the Atlantic meridional overturning circulation (AMOC) at 26°N has now been continuously measured by the RAPID array over the period Apr 2004 - Sept 2018. This record provides unique insight into the variability of the large-scale ocean circulation, previously only measured by sporadic snapshots of basin-wide transports from hydrographic sections. The continuous measurements have unveiled striking variability on timescales of days to a decade, driven largely by wind-forcing, contrasting with previous expectations about a slowly-varying, buoyancy forced large-scale ocean circulation. However, these measurements were primarily observed during a warm state of the Atlantic Multidecadal Variability (AMV) which has been steadily declining since a peak in 2008-2010. In 2013-2015, a period of strong buoyancy-forcing by the atmosphere drove intense watermass transformation in the subpolar North Atlantic and provides a unique opportunity to investigate the response of the large-scale ocean circulation to buoyancy forcing. Modelling studies suggest that the AMOC in the subtropics responds to such events with an increase in overturning transport, after a lag of 3-9 years. At 45°N, observations suggest that the AMOC may already be increasing. Examining 26°N, we find that the AMOC is no longer weakening, though the recent transport is not above the long-term mean. Extending the record backwards in time at 26°N with ocean reanalysis from GloSea5, the transport fluctuations at 26°N are consistent with a 0-2 year lag from those at 45°N, albeit with lower magnitude. Given the short span of time and anticipated delays in the signal from the subpolar to subtropical gyres, it is not yet possible to determine whether the subtropical AMOC strength is recovering nor how the AMOC at 26°N responds to intense buoyancy forcing.

## 1 Introduction

The Atlantic Meridional Overturning Circulation (AMOC) is a large-scale circulation pattern spanning the Atlantic from south to north, transporting warm waters northward and colder waters southward. It drives a large net northward transport of heat, with one petawatt (1 PW = $10^{15}$ W) released to the atmosphere between 26°N and 70°N, impacting the climate in the North Atlantic region (e.g. Srokosz et al., 2012) including surface temperatures, precipitation and sea level (Delworth and Mann, 2000). The deeper limb of the AMOC is isolated from the atmosphere and can store energy and matter for centuries. Changes to the AMOC during the paleoclimate period are thought to explain the abrupt shifts in climate found in paleoclimate records

(e.g., Barber et al., 1999; Ganopolski and Rahmstorf, 2001), and the current generation of coupled climate models predicts a slowing of the AMOC over the present century in response to increasing greenhouse gases (IPCC, 2013).

This widespread interest in the Atlantic circulation led to the installation of the RAPID-MOCHA-WBTS array (hereafter referred to as the RAPID 26°N array) which has now been in operation, making continuous measurements of the large-scale circulation, for more than 15 years (Frajka-Williams et al., 2019). Given its role in climate, the AMOC was previously thought to be slowly varying, on 'climate' timescales (decadal and longer), and so the ocean and climate communities were surprised when the first published data from RAPID 26°N demonstrated large-amplitude variability on sub-annual timescales (Cunningham et al., 2007). Subsequent releases of the data, following the recovery and redeployment of instruments, yielded new insights into seasonal (Kanzow et al., 2010), interannual (McCarthy et al., 2012) variability, and an observed long-term decline of the AMOC at 26°N through 2016 (Smeed et al., 2014; Smeed et al., 2018). One remarkable finding from the RAPID array was the apparent dominance of wind-forcing on the annual cycle as well as the sustained dip in the AMOC strength in 2009-2010 (Roberts et al., 2013; Zhao and Johns, 2014a; Zhao and Johns, 2014b), calling into question the community's prior expectation that the large-scale overturning circulation is primarily driven by buoyancy forcing at high latitudes (Lozier 2010).

The observations to-date have mostly occurred during a warm period of the multidecadal changes in the large-scale North Atlantic as indicated by the Atlantic Multidecadal Variability (AMV, Zhang et al., 2019). While definitions for this index vary, they generally agree that the AMV was positive (warm) during a period spanning the late 1990s, peaking around 2008-2010, then declining towards zero and even negative values (cool) depending on the definition of the AMV used (Frajka-Williams et al., 2017; Zhang et al. 2019). Numerical investigations into the relationship between the AMOC and AMV demonstrate a causal link with the AMOC driving changes in the AMV, where the northward heat transport by the AMOC accumulates in North Atlantic and generates a positive ocean temperature (subsurface and surface) anomaly that is indexed by the AMV (Moat et al., 2019). The decline from a peak in 2008-2010 occurred just prior to a cold anomaly in the subpolar North Atlantic, termed the 'cold blob', and driven partly by intense subpolar heat loss in the winters of 2013/14 and 2014/15 (Duchez et al., 2016; Josey et al., 2018) and also by reduced northward heat transport by the AMOC over a longer period leading up to the cold blob (Bryden et al., 2020). This cold anomaly heralds both a cooler state in the multidecadal variability, but also provides a large-amplitude 'impulse'-like forcing to the large-scale ocean, in a region with known sensitivity of the AMOC (Robson et al., 2014).

While the subpolar AMOC has been observed since 2014 by the OSNAP array (Lozier et al., 2019), the record is as yet too short to compare the overturning and surface forcing both during and prior to the period of intense forcing (2013-15). However, a multi-dataset estimate of the AMOC at 45N indicates broad agreement between the surface forcing and overturning strength, with the overturning responding to the surface forcing with a lag of 5 years and on timescales of 5 years and longer (Desbruyeres et al., 2019). This record of the overturning strength indicates a strong increase in the AMOC at 45N, with the increase notably commencing before the period of strongest surface heat loss.

Here we report on the latest AMOC transport time series at 26°N from April 2004 through the end of August 2018. We give an overview of the variability of the AMOC transport using the complete record, including the seasonal cycle and interannual variability, as well as the contributions of component parts of the circulation (Florida Current/Gulf Stream transport vs meridional Ekman transport vs mid-ocean transports between the Bahamas and Canary Islands). We then update the findings of Smeed et al. (2018) which reported a multiyear reduction in the AMOC strength using changepoint analysis. Based on the RAPID observations and the recent findings at 45°N, we make preliminary investigations into the meridional coherence of the AMOC transport variability between 26°N and 45°N, and the response at 26°N to the impulse forcing in 2013/15. Finally, we place the latest AMOC transport record in context of the larger-scale Atlantic variability, its heat content and the AMV index. These latest results show a possible recovery of the AMOC strength since its lowest point in 2009, but the short duration of the record since 2014 precludes conclusive determination of the AMOC response to buoyancy forcing at this time.

## 2 Data

### 2.1 RAPID 26°N observations and transport calculations

The 14 years of observations at 26°N represent the most complete and longest record of the directly observed AMOC variability currently available. The RAPID array (Fig. 1) spans the middle of the North Atlantic subtropical gyre close to the latitude at which the ocean heat transport is maximum. Here the warm northward flowing waters of the western boundary current are largely confined to the Florida Straits with a small but highly variable part flowing east of the Bahamas in the Antilles Current (Meinen et al., 2019). Across the rest of the section there is a broad southward recirculation of the surface

waters extending across to the coast of Africa where seasonally varying upwelling gives rise to cooler water along the shelf edge. The deep southward flow of the AMOC is predominantly close to the western boundary and transports two distinct water masses: one centered around 1500 m depth, formed within the subpolar gyre, and often referred to as Upper North Atlantic Deep Water (UNADW), and the other below 3000 m originating in the Nordic Seas and referred to as Lower North Atlantic Deep Water (LNADW). Deeper still, Antarctic Bottom Water (AABW) flows northward in the western basin.

The objective of the RAPID array is to obtain a continuous and accurate record of the AMOC volume transport, and the associated meridional heat and freshwater transports.  Here we focus on the volume transport; updated analyses of the heat and freshwater transports will be the subject of a separate study.  There are three principal components to the measurements: (1) the flow through the Florida Straits, the Florida Current, is monitored by a subsea cable calibrated by frequent hydrographic surveys (www.aoml.noaa.gov/phod/floridacurrent/), (2) the flow on the steep continental slope east of the Bahamas is

measured by direct velocity measurements from an array of current meters referred to as the western boundary wedge (WBW), and (3) east of the WBW, geostrophic balance is used to estimate the flow from an array of dynamic height moorings. Instruments include, at present, 155 CTDs (conductivity-temperature-depth), 61 current meters, 3 ADCPs (acoustic Doppler current profilers), an additional 43 CTD-Os (CTDs with oxygen), 36 bottom pressure recorders (BPRs) and 4 PIES (pressure-inverted echo sounders). The dynamic height moorings are arranged in three sub-arrays: the western boundary array, the Mid-

Atlantic Ridge (MAR) array and the eastern boundary array. The use of boundary moorings which sample at high frequency (hourly) enables high frequency (e.g. tidal and mesoscale) variability to be resolved and not aliased (Kanzow et al., 2009). In addition, the ageostrophic meridional Ekman transport is derived from the ERA5 reanalysis for zonal surface stress. A full description of the methodology for calculating the AMOC transports is given in McCarthy et al. (2015), and updated in the dataset release notes at https://www.rapid.ac.uk/rapidmoc/rapid_data/datadl.php.

### 2.2 AMOC transport at 45°N

In order to compare the RAPID AMOC observations to the wider Atlantic, we use an observational estimate of the AMOC at 45°N which uses a combination of satellite altimetry, reanalysis products and in situ ocean data (Desbruyères et al., 2019, after Mercier et al., 2015). Note, however, that the AMOC at 45°N is defined in density classes (AMOCρ). At 26°N, the transport variability is unlikely to be strongly different between the AMOC in depth-space and density-class as isopycnals across the

broad expanse of the basin (6000km) are nearly flat. However, in the subpolar gyre, the overturning is defined in density coordinates (Pickart and Spall 2007; Mercier et al., 2015; Lozier et al., 2019) to better account for the dynamics of buoyancy redistribution in the ocean, which is also carried out by the horizontal gyre circulation. In the subpolar gyre, overturning is a measure of watermass transformation between the northward 'inflow' and southward 'outflow', irrespective of the depth at which it occurs. As the ad hoc reconstruction of the AMOC at 45°N is less constrained than the mooring-based RAPID

estimates at 26°N, a comparison will be used to investigate their potential links.

## 2.3 Other data sets

The sea surface temperature (SST) product used here was the monthly average ERA5 reanalysis at 0.25° resolution (C3S, 2017) from 1979 to present. The winter (January to March) North Atlantic Oscillation (NAO) time series was calculated from the monthly mean NAO from the NOAA Climate prediction centre. The AMV is a measure of the low frequency variability in the Atlantic on multidecadal timescales, calculated from sea surface temperatures (SSTs) as a North Atlantic average, with the background tendency (Enfield et al., 2001) or background field (Trenberth and Shea, 2006) removed. Here, we use the definition following Sutton and Dong (2012) which is the normalized difference between the 10-year smooth Atlantic SST (equator to 65°N, 75°W to 7.5°W) and global mean SST which is close to that of Trenberth and Shea (2006). This definition contrasts from earlier definitions which averaged the North Atlantic SSTs and then detrended over the record. However, detrending is subject to the time period under consideration and does not allow for nonlinear variations in the time series of global SSTs (Frajka-Williams et al., 2017; Zhang et al., 2019).

We also use data from the GloSea5 global ocean and sea ice reanalysis (Blockley et al., 2014; Jackson et al., 2016), which uses the NEMO GO5 ocean model with a nominal resolution of 0.25° and with 75 vertical layers (Megann et al., 2014). It assimilates in-situ and satellite sea surface temperatures; sub-surface ocean profiles of temperature and salinity; sea ice concentration; and sea level anomalies using the NEMOVAR v13 assimilation scheme (Waters et al., 2015). The experiment is described in more detail in Jackson et al. (2016), with a more in-depth comparison to observations and other ocean reanalyses in Jackson et al (2019).

## 3 Methods

### 3.1 Time series processing

The Florida Current transport is produced at daily resolution after a 3-day low-pass filter is applied. Individual instrument records at 26°N are either half-hourly or hourly, and filtered with a 2-day low pass filter to remove tides. Transports are then calculated on a 12-hour grid, with a 10-day low-pass filter applied. Here the data are binned to 10-day time intervals before further analysis. The seasonal cycle is calculated by least-squares fitting an annual and semi-annual harmonic, with a fixed phase and amplitude over the full (2004-2018) record. McCarthy et al. (2015) find that the accuracy of the 10-day binned data is ±1.5 Sv, a figure that was corroborated by the model analysis of Sinha et al. (2018). The accuracy of the mean annual cycle derived from 18 years of data has been estimated using Monte-Carlo technique in which a normal distributed error with standard deviation 1.5 Sv is added to the monthly data. While the annual cycle appears to vary over the record, as noted in Calafat et al. (2018), further investigation of the annual cycle of transports is beyond the scope of the current investigation. Anomalies relative to the seasonal cycle are low-pass filtered using a 540-day Tukey filter.

Spectra are calculated using a Welch's overlapped segment averaging approach, with a Hamming taper and 50% overlap on the detrended, 10-day binned time series. In order to retain variability at low frequencies, while reducing noise at high frequencies, we use three different window lengths following Kanzow et al. (2010).

For investigations into the relationship between the AMOC at 26°N and 45°N, we consider the geostrophic portion of the AMOC transports. i.e., at 26°N, we subtract the Ekman component from the total AMOC. This is because the Ekman component is independently forced at different latitudes and would not be anticipated to show low frequency coherence between latitudes. The AMOC transport at 45°N is computed without a contribution from surface Ekman transport. Both records are then filtered with a 5-year lowpass Tukey filter.

## 3.2 Changepoint analysis

To analyse the variability of the AMOC transports, we use changepoint analysis on the 10-day total AMOC minus Ekman (hereafter AMOC-Ekman) time series. The methodology is described in Beaulieu & Killick (2018) and is similar to that used in Smeed et al. (2018). A suite of eight models were fitted to the data, in which the short term variability is modelled by either random white-noise or a first order autocorrelation [AR(1)] process. The long-term variability is modelled as either a constant value, a linear trend, or one or more changepoints separating periods each linear with time. Combining all these possibilities for both the short-term and long-term variability leads to a total of eight models: (i) a constant mean with a white-noise background, 'Mean', (ii) a constant mean with first-order autocorrelation 'Mean+AR(1)', (iii) a linear trend 'Trend', (iv) a linear trend with first-order autocorrelation 'Trend+AR(1)', (v) multiple changepoints in the mean with a background of white-noise 'Mean+CP', (vi) multiple changepoints in the mean with first-order autocorrelation 'Mean+AR(1)+CP', (vii) multiple changepoints in the trend with white-noise 'Trend+CP', and (viii) multiple changepoints in the trend with first order autocorrelation 'Trend+AR(1)+CP'. For the models with changepoints, we find the number and locations using the pruned exact linear time algorithm (Killick et al., 2012), which performs an exact search considering all options for any possible number of changepoints and select the optimal number/location balancing the overall fit against the length of each segment. The most appropriate model is selected according to the Akaike Information Criterion (AIC). The AIC differences between each model included in the comparison and the model with the smallest AIC are also computed to assess plausibility of all models. As a rule of thumb, a difference larger than 10 indicates that there is essentially no support for a model given the data and the other models at play (Beaulieu & Killick, 2018). To verify sensitivity to the choice of information criterion, the Bayesian Information Criterion for each model is also computed. The changepoint analysis was conducted using the R package EnvCpt (Killick et al., 2018).

## 4 Results

### 4.1 Characterising the variability of the AMOC at 26°N

The AMOC volume transports are given in units of Sverdrups, where $1\ Sv = 10^6\ m^3\ s^{-1}$. To investigate the variability in the AMOC total and component transports, we calculate frequency spectra (Fig. 2). We only consider fluctuations with periods longer than 20 days as the method of calculating the AMOC transport assumes zero net meridional mass transport; this assumption is only valid on timescales longer than about 10-days (Kanzow et al., 2007). For periods shorter than about 60 days, Ekman transport dominates the variability of the AMOC; at other sub-annual periods, the variability is similar among all three components. Broad peaks in the spectra are found at both annual and semi-annual frequencies, particularly for the upper mid-ocean (UMO) transports, however on timescales shorter than 1 year, fluctuations in the UMO and Florida Current transports are anti-correlated (Frajka-Williams et al., 2016). This anti-correlation results in reduced power at the semi-annual frequency in the total AMOC as compared to the UMO. At periods longer than a year, the AMOC variability is dominated by the UMO transport.

In view of the large and broad spectral peaks we have decomposed the time series into three parts: the seasonal cycle, an interannual signal, and the residual high frequency signal (Fig. 3). There is a substantial seasonal cycle with an amplitude of 2.0±0.1 Sv and 0.7±0.1 Sv (mean and standard error from Monte Carlo estimation) for the annual and semi-annual harmonic, explaining 11% and 2% of the variance, respectively. The residual timeseries, likewise, retains substantial variability with a range of 21.6 Sv and a standard deviation of 3.4 Sv. About 20% of the residual variance is associated with the estimated error of ±1.5 Sv for the 10-day binned data. The large amplitude, sub-annual variability is a compelling reason why continuous, time-resolved in situ observations are required to firmly establish the mean value of the AMOC transports.

For the remainder of the paper, we focus on the low frequency (interannual) variability of the AMOC and component transports (Fig. 4). Both from the spectra and the time series in Fig. 4, it is clear that the low frequency variability in the total overturning transports is governed primarily by the mid-ocean transports, i.e., the upper mid-ocean component and the LNADW layer. This is consistent with previous investigations into the AMOC variability, which showed smaller interannual variability in the Ekman and Florida Current transports than the mid-basin (Bahamas to Canary Islands). It is interesting to note, however, that a reduction in the Ekman transport closely follows the two minima in the UMO transport (2009 and 2012).

The low frequency changes in the AMOC are acyclic, and, based on data through 2012, were described using a linear trend by Smeed et al. (2014). However, the tendency of the time series through 2016 was not monotonic (Smeed et al., 2018), rendering a linear trend less useful at describing the observed variability. Instead, a changepoint analysis was used to fit a model to the total AMOC transport, concluding that for the record through 2016, the total AMOC transport variations were best described by two periods with constant mean values, separated by a single changepoint in 2008-2009 (Smeed et al., 2018). Here, we apply an updated version of the changepoint analysis to the AMOC-Ekman time series through 2018 (Fig. 5). This analysis also finds a changepoint in 2008 (Fig. 5b) in accord with the previous result.

Overall, these results are consistent with the previous analyses of the low frequency variability of the AMOC transport and its component parts. However, we note from the table of annual means (Table 1), that the mean in 2017/18 (calculated over the period 1 April 2017 - 31 March 2018) was $17.8 \pm 1.4$ Sv (mean ± standard error, computed on the 10-day binned time series). The standard errors are large, due to substantial sub-annual fluctuations in the AMOC strength. The AMOC transport in the 2017/18 year ($17.8 \pm 1.4$ Sv) is larger than the recent minimum in 2009/10 ($13.5 \pm 1.3$ Sv), but this does not represent a return to the high AMOC transport values near the beginning of the observational record (2005/06, $20.9 \pm 1.2$ Sv). While the interannual time series appears to show a steadily, if weakly, increasing AMOC transport (Fig. 4a), this is not identified as the leading behaviour in the changepoint analysis and so is not yet a statistically significant increasing tendency.

**4.2 AMOC relationship between 26°N and 45°N**

The 2013/14 and 2014/15 winters saw the return of deep convection in the Labrador Sea in two great impulse events (Yashayaev and Loder, 2016). These localised deep convection events are part of wider and longer-term intensification in subpolar water mass transformation following the minimum in 2005 (Desbruyères et al., 2019). While deep convection is not equivalent to water mass transformation (a distinction emphasised by the OSNAP results, Lozier et al., 2019), it is a potential consequence of the continued buoyancy loss in the subpolar gyre. The overall intensification of the light-to-dense water mass transformation rates since 2005 has led to an intensification of the AMOC at the southern exit of the subpolar gyre since 2010, after a delay of 5-6 years, as found in a recent observational analysis (Desbruyères et al, 2019). Building on previous studies, the arrival of such a signal at subtropical latitudes can be anticipated after 3-9 years, based on models (Johnson and Marshall, 2002; Zhang 2007) and observations (Molinari et al., 1998; van Sebille et al. 2011). Lagrangian studies have been used to identify when newly formed dense waters from the subpolar gyre reach the subtropics, with anomalies moving with the currents via advection (e.g., Bower et al., 2009; Zou et al., 2016; Jackson et al., 2016). However, transport time series can also adjust more rapidly through a fast boundary-wave mediated response of lower latitude AMOC variability to high latitudes forcing. Such a response can potentially be identified by lag correlation or coherence analysis of AMOC transport time series, rather than hydrographic anomalies. Based on the increase in subpolar watermass transformation peaking in 2013-2015 and various time lags between the subpolar-to-subtropical AMOC strength determined from numerical simulations, we would anticipate a sign of the increasing subtropical AMOC by 2018-2022. Determining the particular timing of the adjustment would provide critical groundtruth to meridional coherence investigations.

To investigate meridional coherence, we use the AMOC variations at 26°N and 45°N (Fig. 6a). We have removed the ageostrophic Ekman component to isolate AMOC - Ekman as the geostrophic part of the overturning. Ekman transports are

forced independently at each latitude, while the geostrophic part of the overturning is the part of the signal that we would expect to show meridional coherence. The records are short, particularly the in situ observations at 26°N, for the filtering applied (5-years), but both latitudes show a decrease in the AMOC - Ekman over the 2004 - 2011 period of more than 3 Sv (45°N) and 2 Sv (26°N). This is followed by an increase at 45°N commencing around 2010-2011. Due to the length of the filter (5-years) and the relatively short duration of the in situ 26°N observations, we additionally use GloSea5 estimates at 26°N for a longer overlap period (Fig. 6a).

Comparing the AMOC-Ekman strength between altimetry/hydrography observations 45°N and GloSea5 estimates at 26°N, we find that they show similar timing of relative peaks (1996-1997, 2004-2005) and troughs (2000-2001, 2011, 2011-2013). The near coincidental occurrence of peaks and troughs is consistent with an expectation of some meridional coherence between latitudes. Since 2010, the AMOC at 45°N has been increasing. However, at 26°N the AMOC transport does not yet show a significant increase (see Section 3.2).

With the relatively short duration records and the absence of a clear impulse anomaly to track between latitudes, it is not yet possible to identify the timescale of adjustment between the subpolar and subtropical AMOC strength. It appears, however, from comparing the 45°N observational estimate of the AMOC and 26°N from Glosea5, that the adjustment timescale may be short (0-2 years). In contrast, within the GloSea5 reanalysis itself there was a mean lag of 7 years between a peak in Labrador Sea density and the AMOC at 26°N (Jackson et al., 2016). This discrepancy is difficult to reconcile. While GloSea5 has been validated against the 26°N observations, there does not exist an equivalent long AMOC record in the subpolar gyre to verify GloSea5: the OSNAP estimate of the AMOC is too short (21 months) to verify interannual variability of reanalyses (Lozier et al., 2019) and the method used at 45°N with altimetry and gridded hydrography may be subject to errors particular in resolving higher frequency anomalies at the boundary.

It is further worth noting that the AMOC at 45°N is in density space, following the choice in Desbruyères et al. (2019); the AMOC$_z$ at 45°N is in phase with the AMOC$\rho$, but with lower amplitude (Desbruyères et al., 2019, Figure S4). In addition, the ratio of meridional heat transport to AMOC, a measure of how 'efficient' the overturning circulation is at fluxing heat, is greater at 26°N than 45°N (Johns et al., 2011; Desbruyeres et al., 2019). This means that smaller amplitude fluctuations of the AMOC 26°N than 45°N may be associated with equivalent heat transport variability. More thorough investigations into depth- and zonal-distribution of changes at 26°N that accompany the subtle intensification of the overturning strength are pending. These may enable a more conclusive determination of the arrival of the buoyancy-forced signals in the subtropical North Atlantic.

### 4.3 Ongoing changes in the wider Atlantic

To place the low frequency variability of the AMOC noted above in the wider Atlantic context, we consider large-scale variations in SST and atmospheric variability. On the one hand, the AMOC is anticipated to respond to wind- and buoyancy-forcing, and on the other, it drives heat transport and through it, heat content and SST changes. On multidecadal timescales, Gulev et al. (2013) provided observational evidence that in the mid-latitude North Atlantic and on timescales longer than 10 years, surface turbulent heat fluxes are indeed driven by the ocean and may force the atmosphere, whereas on shorter timescales the converse is true. Numerical simulations identified a driving role in the subtropical meridional heat transport for temperature tendencies in the subpolar North Atlantic (Moat et al., 2019). While the current record of in situ observations is too short to fully-investigate multi-decadal relationships, we can look more closely at the period of the observations and the longer records of SST to evaluate whether the observed variations in the Atlantic, as indexed by the AMV, follow the patterns predicted by the numerical simulations.

The AMV is a record of the multidecadal variations in the North Atlantic, based on SST (Fig. 6b). During the period prior to 2007/2008 the AMOC is generally in a positive state (Fig. 6a), which leads to greater than average northwards heat transport

as the AMOC volume transport and meridional heat transport are proportional (Johns et al., 2011). This northward heat transport then leads to a warming North Atlantic, consistent with a positive AMV state (Fig 6b - from increased SST). After 2007/2008 the AMOC moves into a negative state with less than average northwards heat transport, which is followed by decreasing SST's and reducing AMV. Using a coupled climate model Moat et al., (2019) showed that on decadal time scales changes the AMOC leads the AMV by about 5 years. There is evidence here to suggest that the AMV does not respond instantaneously with the AMOC and the AMOC may lead the AMV. However, the length of the AMOC at 26°N is currently too short for the lagged correlations to be statistically significant.

The long-timescale fluctuations in the AMV contrast with atmospheric variability, as measured by the North Atlantic Oscillation index (NAO) which tends to vary on shorter 3-5 year timescales. The low-passed NAO was in a positive state with a maximum around 1990 and declining to near zero in 2005. During this period AMOC was in a positive state moving more than average heat northwards (GloSea5, Fig 6a). As the NAO declines into a -ve state there is a reduction in the surface heat loss in the subpolar region of the North Atlantic, which is followed by a reducing AMOC strength. Since 2010 the NAO is recovering from a minimum and moving towards a NAO+ state, resulting in enhanced heat loss in the subpolar North Atlantic and strengthening the AMOC. Given the lag between the AMOC and AMV described above we would anticipate an increase in the AMV with increasing AMOC, which is consistent with the hypothesis illustrated in Sutton et al. (2018).

From this large-scale view of the Atlantic, we can conclude that the observed and simulated AMOC variability (using 14 years of RAPID observations and GloSea5 reanalysis), SST variability (indexed by the AMV) and atmospheric forcing (captured by the NAO) are consistent with other studies (Moat et al., 2019; Sutton et al., 2018). A positive NAO period is associated with stronger heat loss from the subpolar North Atlantic, providing buoyancy forcing to strengthen the AMOC. And a strong AMOC will transport more heat northward leading to a warmer North Atlantic (more positive AMV). While the recent decade offers a change in state of the Atlantic (AMV) as well as anomalous buoyancy forcing in subpolar North Atlantic (2013-2015), the time series of directly-measured AMOC variability at 26N is not yet long enough to conclusively test the mechanisms linking buoyancy forcing to circulation change, and leading to changes in ocean heat content. A more complete diagnosis of the short-term heat budget (2014-2020) and the relative contributions of ocean transports and surface fluxes is beyond the scope of this paper, but currently underway.

## 5 Conclusions

From the nearly 15-year long record of the AMOC variability at 26°N, we can characterise the transports as highly variable on all timescales, with high frequency variability (shorter than 60 days) dominated by rapid fluctuations in the zonal winds across 26°N, seasonal cycles contributed to by the UMO transport between the Bahamas and Canary Islands, and low frequency variability dominated by the UMO transports and mirrored in the LNADW layer (3000-5000m). This is in agreement with previous investigations into the seasonal cycle (Kanzow et al., 2010; Duchez et al., 2014), high frequency variability (Moat et al., 2016) and interannual variability (McCarthy et al., 2012), compensation between components (Kanzow et al., 2007; Frajka-Williams et al., 2016). Using the full duration of the record, we further investigate the tendency in the record finding that the decline previously identified as a trend (Smeed et al., 2014) and as a changepoint between two periods with a higher and lower mean (Smeed et al. 2018) has not yet reversed. While the lowpass filtered AMOC time series appears to show an increasing tendency since 2009 (Fig. 3c), this increase is not statistically significant.

The recent intense heat loss in the subpolar North Atlantic (2013-2015) and the extension of the RAPID record through 2018, motivated an investigation into when and how the RAPID transports would respond to buoyancy forcing in the subpolar gyre forcing. In situ estimates of the overturning at 45°N indicate that at 45°N, near the southern boundary of the subpolar gyre, the overturning strength is already intensifying following sustained buoyancy forcing in the subpolar gyre (Desbruyères et al. 2019). Comparing the transport variability at 26°N and 45°N, we show some indication of a potential lead-lag relationship

(45°N leading changes at 26°N by 0-2 years) in the AMOC-Ekman transports, but with stronger amplitude variations at 45°N. As yet, however, the available AMOC time series at 26°N does not show a statistically significant increase since the low period in 2010 (Fig 5).

In addition to the AMOC responding to subpolar changes, it is anticipated to cause change in the northern North Atlantic through changes in the meridional heat transport of the AMOC. The phase-relationship identified in the modelling study of
325 Moat et al. (2019) relies on identifying periods where the AMOC is increasing or decreasing, or where it is positive vs negative (corresponding to increasing or decreasing accumulated northward heat transport). While the in situ record at 26°N is too short to conclusively determine the lag, a comparison between model reanalysis (GloSea5) AMOC at 26°N and the AMOC at 45°N supports this timing. Using these longer records, we find that the changes in the AMOC strength are consistent with an ocean role in driving variations in North Atlantic temperatures, but a more complete heat budget analysis is under investigation for
a conclusive determination of the relative importance of ocean transports vs surface forcing.

The transport time series at 26°N in the Atlantic of the large-scale ocean circulation has yielded new insights into the variability of the overturning circulation (Srokosz and Bryden, 2015). The results here extend our knowledge of the AMOC variability through 2018, finding that the AMOC is marginally stronger in the period 2014-2018 than the preceding period (2009-2014) using a changepoint analysis. However, the lead-lag relationships between the AMOC at two latitudes (26N and 45N) cannot
be conclusively determined. Additionally, the AMOC at 26N does not yet appear to be responding to the intense buoyancy loss in the subpolar gyre in 2013-2015. Based on the findings in Desbruyères et al. (2019), that the AMOC at 45°N lags basin-wide surface-forced transformation in the subpolar gyre by 5 years, and the tentative 0-2 year lag from the AMOC at 45°N to the AMOC at 26°N, we would anticipate an intensification in the overturning strength at 26°N in response to the 2013-15 forcing by 2018-2022, and may become apparent in the next recovery of the RAPID observations.

**Data availability**

The RAPID-MOCHA-WBTS time series (Smeed et al., 2019) is available at http://www.rapid.ac.uk/rapidmoc. ERA5 sea surface temperature (SST) is available via https://www.ecmwf.int/en/forecasts/datasets/reanalysis-datasets/era5. The GloSea5 time series is available from (Jackson et al., 2019). The 45°N time series of Desbruyères et al., (2019) is available from the author on request.

**Author Contributions**

EFW, DAS, BIM, CB, DD wrote the manuscript with input from all authors. BIM, DR, DD, DAS, HLB contributed to the transport calculations. DAS, CB, BIM, EFW, ASF, LCJ performed the analysis. WEJ, DV, MOB, BIM, DAS, DR, EFW, HLB contributed to the data collection.

**Competing interests**

The authors declare they have no conflict of interest.

**Acknowledgements**

This research was supported by grants from the UK Natural Environment Research Council for the RAPID-AMOC program and the ACSIS program (NE/N018044/1), by the U.S. National Science Foundation (grant 1332978), by the U.S. National Oceanic and Atmospheric Administration (NOAA) Climate Program Office (100007298), and by the U.S. NOAA Atlantic

Oceanographic and Meteorological Laboratory. The authors thank the many officers, crews, and technicians who helped to collect these data.

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

|  | AMOC (Sv) | Ekman (Sv) | Florida Current (Sv) | UMO (Sv) |
|---|---|---|---|---|
| 2004/05 | 18.4 ± 4.7 | 3.9 ± 3.7 | 32.0 ± 3.0 | -17.5 ± 2.6 |
| 2005/06 | 20.9 ± 4.0 | 4.4 ± 2.5 | 32.0 ± 2.4 | -15.5 ± 2.6 |
| 2006/07 | 20.3 ± 3.3 | 5.1 ± 2.9 | 31.6 ± 1.9 | -16.3 ± 2.8 |
| 2007/08 | 18.9 ± 3.5 | 4.9 ± 2.7 | 31.7 ± 2.4 | -17.6 ± 2.6 |
| 2008/09 | 18.0 ± 3.4 | 5.3 ± 2.8 | 31.6 ± 3.6 | -18.7 ± 3.8 |
| 2009/10 | 13.5 ± 4.4 | 3.1 ± 3.9 | 30.7 ± 2.5 | -20.2 ± 2.5 |
| 2010/11 | 17.4 ± 4.0 | 4.1 ± 3.4 | 31.1 ± 2.9 | -17.6 ± 3.7 |
| 2011/12 | 18.0 ± 2.9 | 5.8 ± 2.6 | 31.1 ± 2.3 | -18.7 ± 2.9 |
| 2012/13 | 14.8 ± 4.4 | 3.8 ± 3.5 | 30.8 ± 3.0 | -19.6 ± 2.8 |
| 2013/14 | 18.0 ± 3.0 | 5.7 ± 2.6 | 31.5 ± 2.9 | -19.0 ± 3.3 |
| 2014/15 | 17.2 ± 2.9 | 5.1 ± 2.6 | 30.4 ± 2.6 | -18.2 ± 2.5 |
| 2015/16 | 17.5 ± 3.6 | 4.7 ± 2.8 | 31.6 ± 3.0 | -18.8 ± 3.3 |
| 2016/17 | 18.0 ± 3.7 | 5.0 ± 2.7 | 32.4 ± 3.6 | -19.4 ± 3.9 |
| 2017/18 | 17.8 ± 4.9 | 5.1 ± 3.7 | 30.7 ± 2.3 | -17.9 ± 3.1 |

**Table 1**. The annual means of the AMOC volume transport and components in Sverdrups (1 Sv = $10^6$ m$^3$ s$^{-1}$). Values are given as the annual mean ± the standard deviation of the 10-day binned values for that year. Annual means are computed from 1 April through 31 March. Positive values indicate northward transport, while negative values are southward. The de-correlation time is of the order of 20-30 days for all variables, and so the standard error is about square root (1/12) multiplied by the
standard deviation. The de-correlation time is 20-35 days for all variables, and so the standard error is between $\sqrt{(1/18)}$ and $\sqrt{(1/10)}$ multiplied by the standard deviation.

| Model | AIC | AIC differences | BIC |
|---|---|---|---|
| Mean | 2296.5 | 250.9 | 2304.8 |
| Mean + CP | 2193.0 | 147.5 | 2213.8 |
| Mean + AR(1) | 2082.9 | 37.4 | 2095.3 |
| Mean + AR(1) + CP | 2045.5* | 0.00* | 2074.6* |
| Trend | 2255.3 | 209.8 | 2267.8 |
| Trend + CP | 2175.8 | 130.3 | 2204.9 |
| Trend + AR(1) | 2068.3 | 22.8 | 2085.0 |
| Trend + AR(1) + CP | 2068.3 | 22.8 | 2085.0 |

**Table 2**. Comparison of the eight models fitted to the AMOC-Ekman time series. The Akaike Information Criterion (AIC)
and Bayesian Information Criterion (BIC) obtained for each model are presented. The most appropriate model from these
information criterion is selected as the smallest and highlighted with a *. The AIC differences between each model fitted and
the "best model" (with the smallest AIC) are also presented. The differences are all large (>10), indicating that there is no
other model amongst those compared that fits the data reasonably well. Note that because no changepoints were detected under
the Trend + AR(1) + CP model, the AIC and BIC are the same as the Trend + AR(1) model.

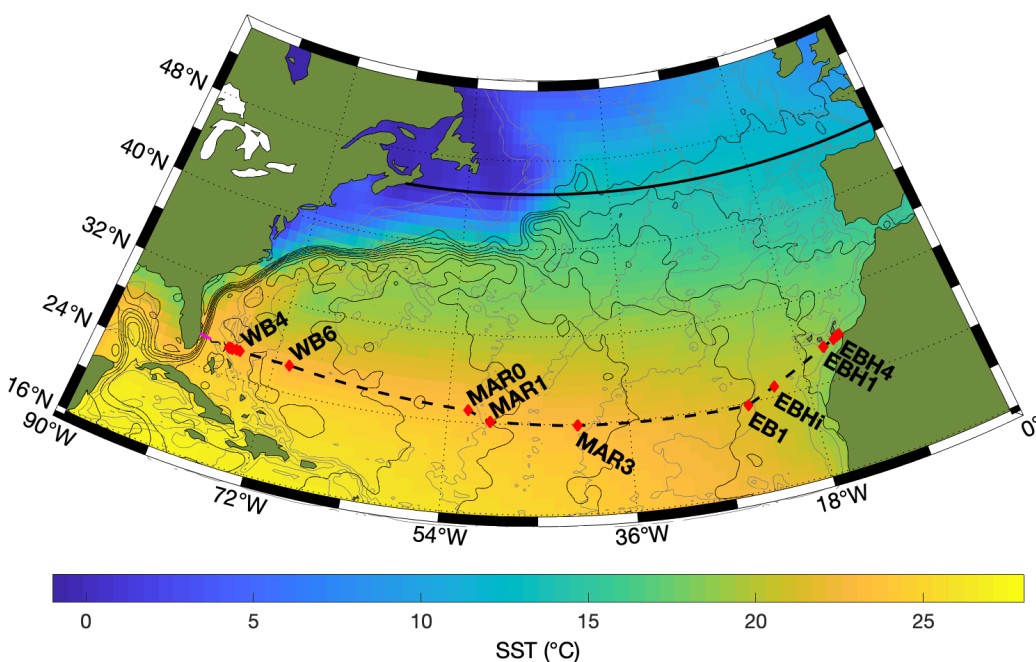

**Figure 1** The RAPID 26°N array traverses the subtropical gyre of the North Atlantic. The magenta line shows the location of the subsea cable in the Florida Strait and red diamonds connected by a dashed black line show the location of moorings. 'WB', 'MAR', and 'EB' denote, respectively, moorings in the western boundary, mid-Atlantic Ridge and eastern boundary sub-arrays. For clarity, not all moorings are labelled. The colour shows mean sea surface temperature (SST) in March (average of 1999 to 2018) and the continuous black lines are the corresponding contours of sea surface height (contour interval 0.1m). Contours of water depth at 1000, 3000 and 5000 m are shown in grey. The thick black line at 45°N indicates where multiple data sources have been used to estimate the AMOC at the boundary between the subtropical and subpolar gyres (Desbruyères et al., 2019).

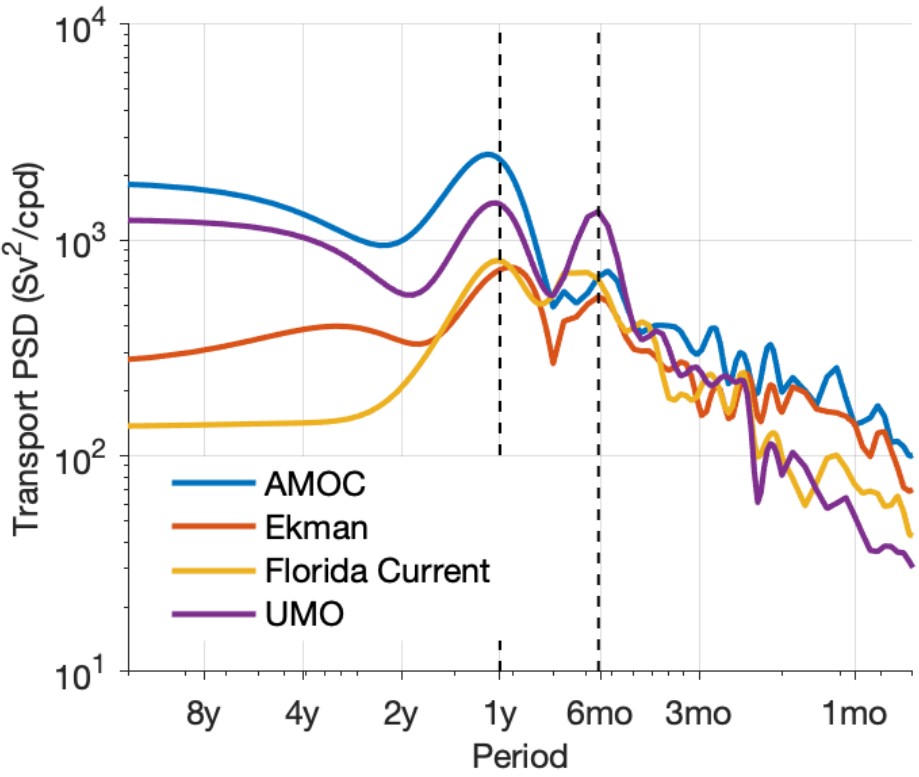

**Figure 2** Power spectral density of the AMOC and its component parts as a function of period. The vertical dashed lines highlight the annual and semiannual frequencies.

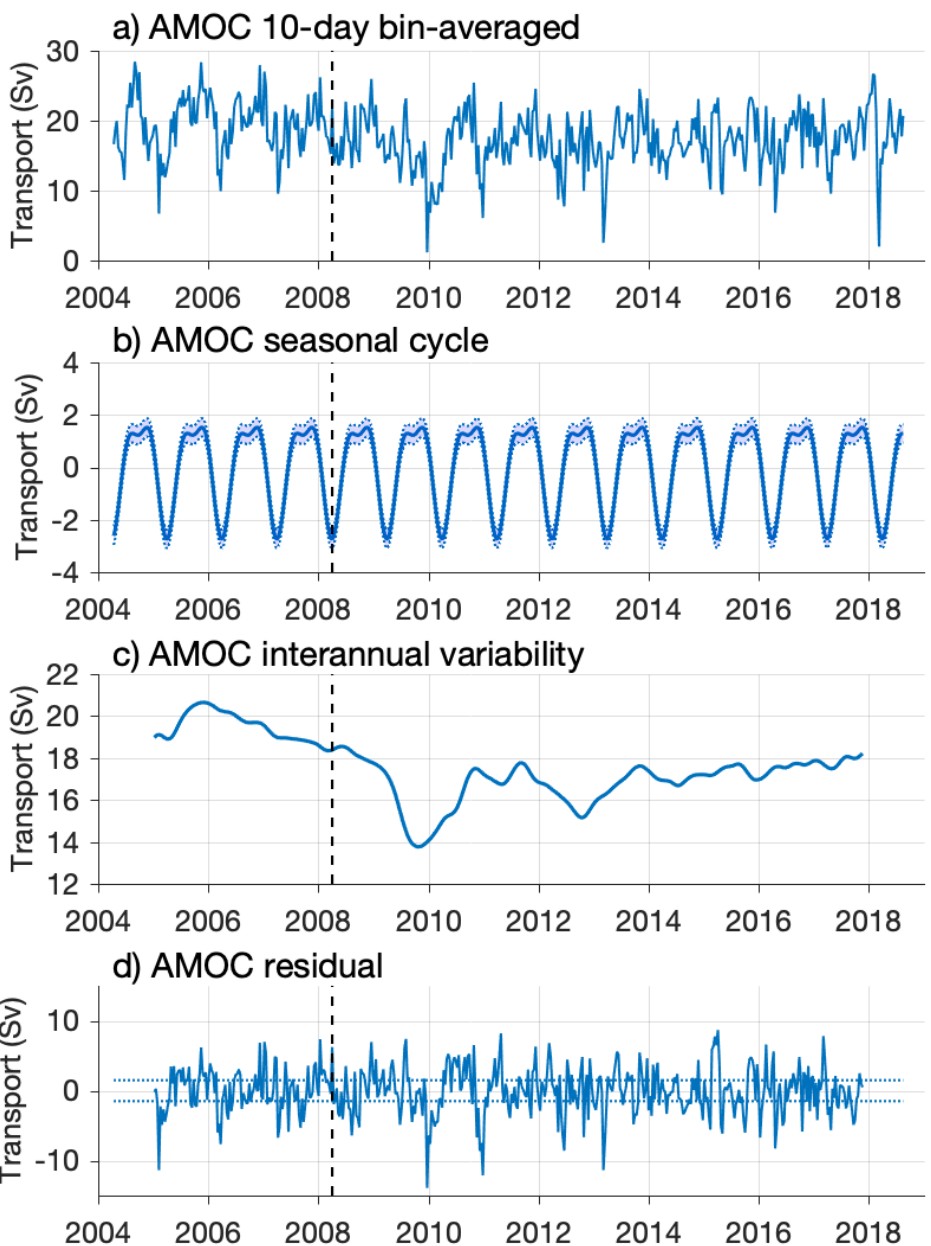

**Figure 3** The total AMOC at 10-day resolution (a), can be decomposed into a seasonal cycle (b), interannual variability (c), and a residual (d). The interannual component is obtained by filtering the data with a 540-day low-pass filter after removal of the mean seasonal cycle. In (b) the dotted lines show the annual cycle ± one standard error, and the dotted lines in (d) are ±1.5 Sv the estimated error of 10-day binned data.

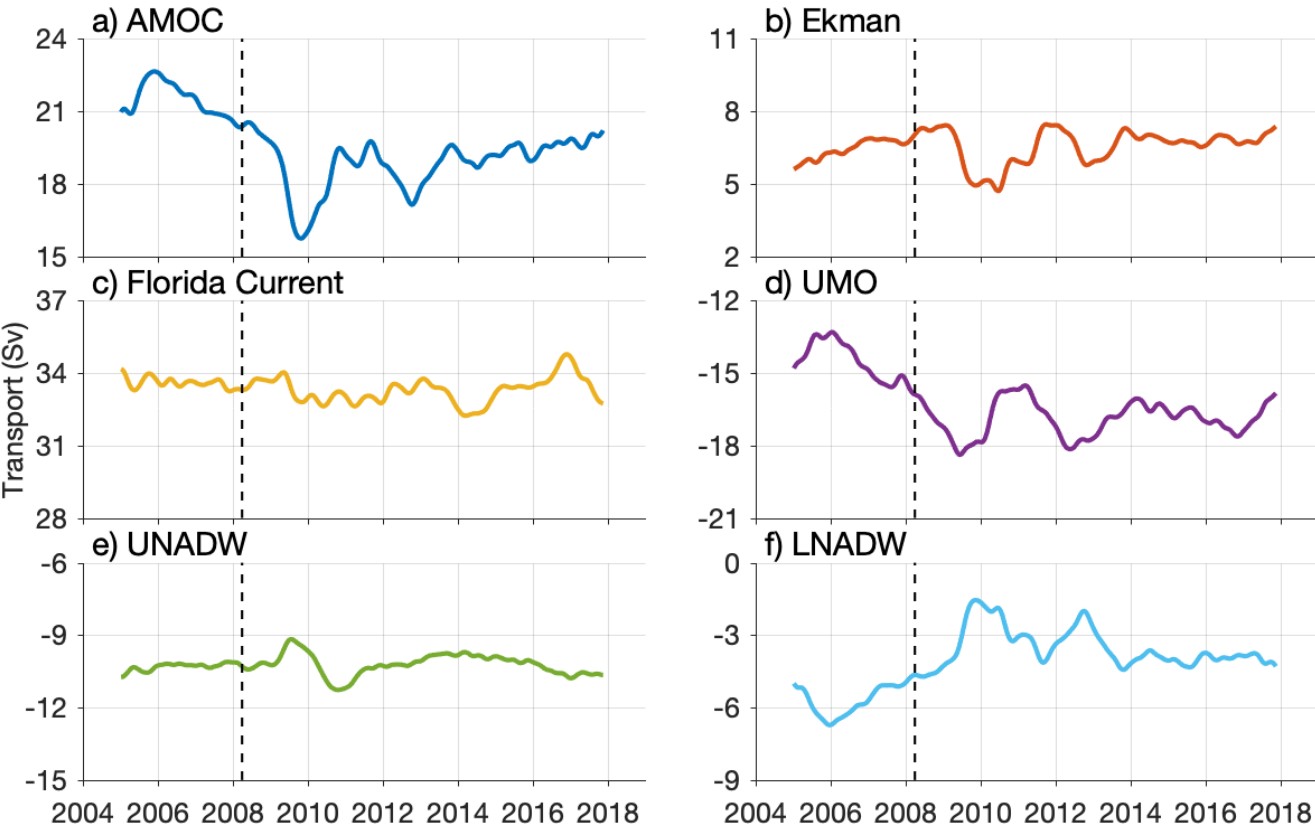

**Figure 4** Interannual variability of the AMOC at 26°N and its component parts: (a) AMOC, (b) Ekman, (c) Florida Current, (d) Upper mid-ocean (UMO), (e) Upper North Atlantic Deep Water (UNADW), and (f) Lower North Atlantic Deep Water (LNDW).

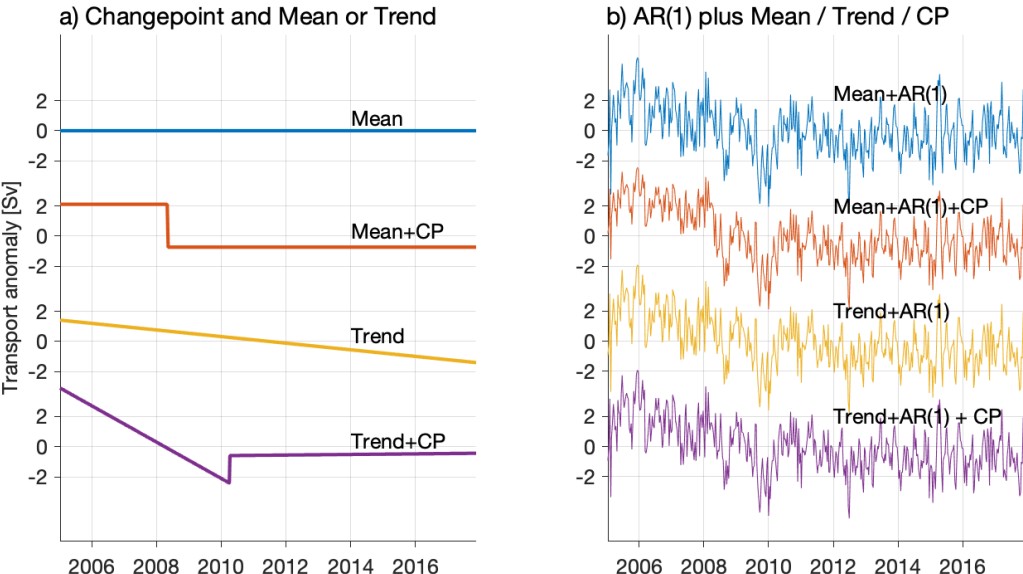

**Figure 5** Changepoint analysis of the AMOC-Ekman time series. In panel (a), only a mean or a trend, with or without a changepoint are fit. In panel (b), an AR(1) is also fit. The model with the best overall fit is the Mean + AR(1) + CP model (red, right) according to the AIC (see Table 2), indicating that the time series can best be explained by an AR(1) time series with a change in the mean in 2008.

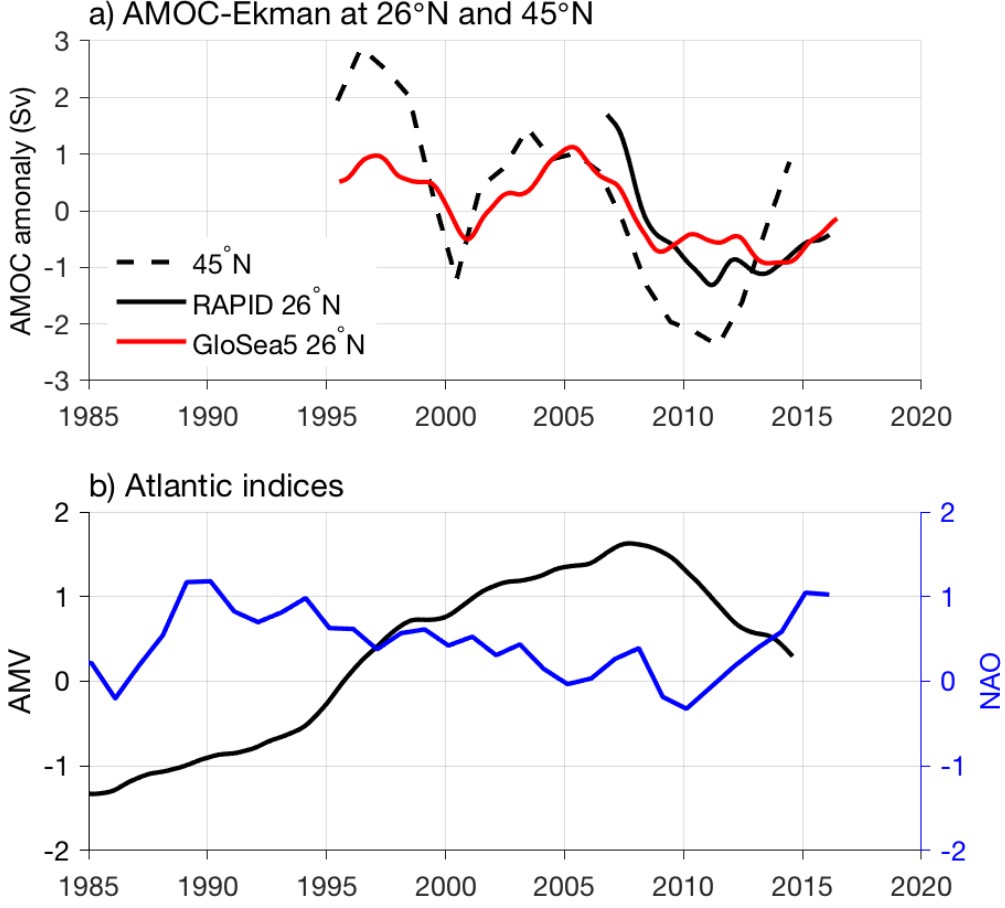

**Figure 6 (a)** AMOC anomalies from RAPID at 26°N (black, Sv), 26°N GloSea5 reanalysis (red, Sv), AMOC 45°N (black dashed, Sv). b) The AMV (black) and NAO (blue). The AMV has been decadally low-pass filtered, with a 5-year low-pass filter applied to the NAO time series. The Ekman transport has been removed from the AMOC time series.