# Peer review of "Pending recovery in the strength of the meridional overturning circulation at $26^{\circ}N$"

_Ocean Science, 2019_

## Referee Comment (RC1) · Anonymous Referee #1 · 25 Feb 2020

Re: "Pending recovery in the strength of the meridional overturning circulation at 26N" by Moat et al.

The authors reviewed the AMOC variability at 26N based on the extended time series from the RAPID-MOCHA array and analyzed concurrent multi-year changes in the North Atlantic. The authors also explored the potential of an intensification of the AMOC at 26N in response to the 2013-2015 strong cooling event over the entire subpolar gyre. The central questions the authors are trying to address are very interesting and critical. But I have several concerns which are outlined below. Mainly, the manuscript presents some speculations of the AMOC changes and their broader impact that seem to be exclusively based on modeled lead-lag relationships between key physical progress derived. How robust those relationships are and over what time

scales are still open questions and can vary considerably across models. And a discussion on how that may affect the analysis is missing. My overall recommendation is that the authors include more details on how those relationships are potentially achieved (AMOC at 26N and 45N, and AMOC-AMV-subplarOHC, etc.) and discuss thoroughly the limitations of modeled relationships and how they might affect the interpretation of the observed changes.

Main concerns:

(1). Subpolar water-mass transformation and AMOC connectivity between 45N and 26N.

The anticipated intensification of the AMOC at 26N appears to be relying strongly on (a) an ongoing AMOC intensification at 45N following the basin-wide strong cooling in the subpolar region, and (b) a subpolar-subtropical AMOC connectivity. However, I found the evidence for those two conditions (or assumptions) to be not sufficient if not inadequate.

If a strengthening of the subpolar water mass transformation leads an increasing AMOC at 45N by 5-6 years (line 193 in this manuscript). Then why did the AMOC at 45N already begin to increase around 2011 (Figure 6a)? Also, of note is that Desbruyères et al. (2019) assumed an immediate export of the newly formed dense waters without any accumulations of water volumes. Recent Lagrangian studies, however, show much longer time scales (> 10 years) for those dense waters to be exported to the subtropics (e.g., Jackson et al. 2016; Zou et al. 2016). A more comprehensive discussion will be needed to reconcile those different perspectives on how the subpolar water mass transformation may impact overturning variability.

The authors then suggested that a larger AMOC at 45N leads a larger AMOC at 26N by 0-2 years. But using the same Glosea5, Jackson et al. (2016) suggested that the AMOC anomalies at 45N precedes 26N by about 10 years. How to reconcile such a significant discrepancy? Is it related to the use of the observed AMOC at 45N and the

OSD
modeled AMOC at 26N? It isn't clear why the authors used Glosea5 other than any other models/reanalysis. Some validations of Glosea5 will be needed, in particular, on how well Glosea5 reproduces the AMOC variability and connectivity. In addition, the authors cited Zou et al. (2019) on the connection between the subpolar UNADW and subtropical LNADW transport anomalies. Note that Zou et al. (2019) suggested that such a latitudinal AMOC connection can be due to gyre-dependent forcing; only strong LNADW transport anomalies can propagate southward from the subpolar region to the subtropics in 4 years. A discussion on this and a reconciliation with the presented analysis are currently missing in the manuscript.

(2). AMOC-AMV-subpolar OHC relationships.

It is a bit confusing about the relationships of AMOC-AMV-subpolar OHC. The authors first suggested that the AMOC lead the AMV by  $\sim$ 4 years as shown in a high-resolution model (Moat et al. 2019), but then pointed out the AMOC maximum at 45N precedes the AMV by  $\sim$ 10 years in Glosea5 (lines 243-244 in this manuscript). Does it imply that the AMOC-AMV relationship is just model-specific? In addition, how does the AMOC-AMV relationship relate to the subpolar OHC changes?

The authors suggested a relationship between the weakened heat transport in the subtropics (i.e., in relation to a weak AMOC state) and the cooling subpolar gyre during 2013-2015. Should it be focused on the heat transport at 45N that is at the southern boundary of the subpolar gyre? The AMOC at 45N appears to be strengthening after 2011 (Figure 6a), indicating an increasing northward heat transport during the cooling period. How to exclude the impact from the strengthened atmospheric forcing during 2013-2015 (e.g., de Jong and de Steur 2016)? My suggestion is to add a time series of the surface heat flux over the subpolar region during the overlapping period of 1985-2018 and discuss accordingly their potential impact on the oceanic changes. Otherwise, in my opinion, it is hard to draw any conclusions on how the AMOC changes lead the changes in the subpolar OHC. OSD
Other comments:

Lines 162-163: To utilize the lengthy record, the authors could put error bars on the monthly values and comment on how robust the seasonal cycles are.

Lines 180-181: Need more information on Figure 5. How to understand the different change points defined by Mean+CP and Trend+CP? Mean+CP shows an earlier change point around 2008. Also, it is not clear from Figure 5 why Mean+AR(1)+CP is the overall best fit. Please add more details on how this was determined.

Line 184: The standard deviation clearly varies with the time scales over which it is derived. I would suggest the authors show the standard error in the mean instead, which seems to be more helpful when determining how distinct the time-mean transports are between two years or any two periods.

Line 189: Is section 4.2 just about the relationship to 45N? If so, better to be more specific.

Lines 190-203: Please see my main concern #1.

Line 208: Is the timing of the AMOC increase at 45N (2010-2011) sensitive to the size of the filter?

Lines 213-214: Is the difference in the variability the same between the AMOC at 45N and 26N both in Glosea5?

Line 238: The authors appear to emphasize a 4-year time lead by the AMOC. But I couldn't find any observational evidence even in this analysis for such a time lead.

Lines 253-255: Please see my main concern #2.

Additional references:

de Jong, M. F., and L. de Steur, 2016, Strong winter cooling over the Irminger Sea in winter 2014–2015, exceptional deep convection, and the emergence of anomalously

OSD
low SST, Geophys. Res. Lett., 43, doi:10.1002/2016GL069596.

Zou, S., and M.S. Lozier, 2016, Breaking the Linkage Between Labrador Sea Water Production and Its Advective Export to the Subtropical Gyre. J. Phys. Oceanogr., 46, 2169–2182, https://doi.org/10.1175/JPO-D-15-0210.1

---

## Referee Comment (RC2) · Anonymous Referee #2 · 9 Mar 2020

Review of Moat et al. "Pending recovery in the strength of the meridional overturning circulation at 26°N", submitted to Ocean Science. The paper presents the latest data from the RAPID-MOCHA array, extending the time series for the overturning circulation at 26N to over 14 years. The data are analysed spectrally, and using techniques such as change point analysis and autoregressive modelling. The time series is compared to the overturning based on GloSea5 reanalysis at the equivalent location and 45N, as well as the AMV signal in sea surface temperature, to place observed changes in the context of our current understanding of Atlantic circulation. The authors find that the expected increase in the overturning signal at 26N cannot be detected by the RAPID observations at this stage.

These are important results, and it is crucial that the up-to-date RAPID time series is

published and made available to the ocean and climate communities. I found the paper clearly written with nice, appropriate figures. The time series analysis techniques (CP and AR) were both innovative and insightful. However, I have some issues with the central scientific focus of the paper and felt that the general thrust of the argument was often misguided. I therefore think significant re-writing is required before publication.

Major comments

1. My overall impression from this paper was that the authors tried, retrospectively, to formulate a compelling question such that the updated RAPID time series is the answer. I am not sure that this sort of hypothesis driven structure is really required when the main purpose of the paper is to present the new data (the value of which cannot be overstated). Besides, the attempt to fit the results into a wider narrative on North Atlantic circulation does not really showcase the value of the data since the RAPID time series neither contradicts nor substantiates the working hypothesis. Comparison with climate indices such as the AMV and NAO is appropriate but does not warrant being the central focus, given the relatively short timescale and inconclusive findings.

2. I think the GloSea5 data is overused and overly trusted to give a realistic representation of the ocean. The authors attempt to reconcile the results with the 45N time series from Debruyeres et al. (2019), but I think too much respect is paid towards these results which are not of comparable stature. It is a worthy discussion point, but the authors need to make it clear that the overturning at 45N remains poorly constrained compared to 26N. I am unconvinced, from figure 6a, that GloSea5 does a sufficiently good job at 26N given the small number of degrees of freedom in the filtered time series. This then calls into question how well the product is likely to capture the meridional connectivity of the two sections (which underpins much of the argument). Are there other reanalysis products and/or models which corroborate the lead/lag relationships from GloSea5? I feel it's needed.

In broader terms, the writing could do more to promote RAPID as the principal source

of our understanding of the overturning circulation, rather than another piece of the puzzle. The presentation of GloSea5 and Desbruyeres et al. (2019) as comparable data sources serves only to undermine the RAPID project itself.

Minor Comments

Line 27: "Comparing the two latitudes, the AMOC at 26°N is higher than its previous low" this sentence needs to better distinguish spatial and temporal changes. Line 35: Slightly clumsy sentence, repetition of "on" Line 71: "Guided by". This language ties directly into point 2. RAPID should lead, not follow. Line 83: missing "to" Line 89: The heat and freshwater fluxes are mentioned here but neither shown nor discussed. Perhaps a sentence explaining why? Line 95: Are the CTD-Os a subset of the CTDs? Line 107: "net the" Line 109: GloSea5 should be mentioned here. Line 132: missing "use" Line 150: Should say "Results" Line 158: why is "anti-correlated" repeated inside the brackets? Line 158: Clarify: there is no correlation information in the spectral plot. Line 169: Use the LNADW acronym Line 172: "that a reductions" Lin3 186: This sentence could be improved, just state the maximum and minimum values/times for comparison. Line 210: All this tells us is that GloSea5 is dynamically consistent with itself. It could still be wrong. I assume GloSea5 changes are forced by Lab sea deep convection, which we know many models get wrong, even if it does assimilate observations. Line 218: An excellent point, and an example of where it is appropriate to turn to reanalysis. Line 232: You can, but I think this analysis seems uncoupled from the RAPID results. Line 256: Be quantitative. What is the minimum fraction of the mean? Line 266: This is the first mention of the 34.5S array (in the conclusions). Line 272: Insert "within a reanalysis framework". Line 288: Perhaps substitute "understanding" for "knowledge". Figure 1: Red text on green very hard to read for colour blind people.

---

## Author Comment (AC1) · 15 May 2020

Pending recovery in the strength of the meridional overturning circulation at 26°N Ben. I. Moat, David. A. Smeed, Eleanor Frajka-Williams, Damien G. Desbruyères, Claudie Beaulieu, William E. Johns, Darren Rayner, Alejandra Sanchez-Franks, Molly O. Baringer, Denis Volkov, Laura C. Jackson, Harry L. Bryden

We thank the reviewers for their time in commenting on this paper. We have prepared a detailed response to reviewers #1 and #2.

Reviewer 1 notes that we are relying on the expectations that (a) AMOC transport will increase as a result of strong buoyancy forcing in the subpolar North Atlantic and (b) that there is some relationship between the AMOC in the subpolar and subtropical

gyres. These are indeed assumptions that we are working with as they are the prevailing view of the AMOC circulation variability on long timescales. On shorter timescales, the transport variability is confounded by higher frequency/shorter period fluctuations that are wind driven. This short timescale variability presents significant challenges in identifying meridional connectivity, particularly when time series are themselves short.

We are further relying on the assumption that meridional coherence of the circulation, if it exists, will appear in transport fluctuations (e.g., Zhang 2010, Bingham and Hughes, 2009) rather than in watermass advection (e.g., Zou et al. 2016). The arrival of watermass signatures, while easier to identify in longer hydrographic records, is a complicated integral of the transport variability along spreading paths, and represents a complementary measure of ocean circulation change.

We have attempted to identify transport covariance between the best available AMOC observations at 26N, and the longest available estimates at 45N. To extend the time-series at 26N, we have used the GloSea5 reanalysis which was shown to capture the interannual variability of the RAPID array at 26N albeit with reduced amplitude (Jackson et al. 2019). Using these records, we still cannot conclude a definitive lead-lag relationship between two latitudes. However, we anticipate that the strong subpolar cooling in 2013-2015 may provide an impulse response that will generate a signal in meridional connectivity above the background high frequency 'noise'.

The GloSea5 time series has been extended to the end of 2018 by Laura Jackson at the UK Met Office, and is now included in this paper (red line Figure 6a). Laura has also contributed to the analysis in the updated paper so we have included her as a co-author.

Anonymous referee #1

Detailed responses to particular points follow: 1) If a strengthening of the subpolar water mass transformation leads an increasing AMOC at 45N by 5-6 years (line 193 in this manuscript). Then why did the AMOC at 45N already begin to increase around

2011 (Figure 6a)?

Desbruyeres et al. (2019) and the AMOC time series at 45N show an increase from a relative minimum in 2010. Similarly, the watermass transformation shows an increase from a relative minimum in 2005 (5 years earlier). However, the watermass transformation due to oceanic heat loss was not significantly greater than zero until 2010. We have updated the text in the manuscript to read:

"These localised deep convection events are part of wider and longer-term intensification in subpolar water mass transformation that was at a minimum in 2005"

2) Recent Lagrangian studies, however, show much longer time scales (> 10 years) for those dense waters to be exported to the subtropics (e.g., Jackson et al. 2016; Zou et al. 2016). A more comprehensive discussion will be needed to reconcile those different perspectives on how the subpolar water mass transformation may impact overturning variability.

Lagrangian approaches identify advective pathways between the subpolar and subtropical regions in the Atlantic, but are not ideal to capture faster boundary wave-mediated changes in transport. Indeed, Zou et al (2016) comments on this issue after finding that the Lagrangian approach did not show any relationship between watermass formation and transport variability. To get around this, they used e-floats on either side of key latitudes to match the transport anomaly signatures, finding that it propagated much more quickly than water parcels did (2 year time lag).

We have added text to emphasize that with our transport time series we are looking at how anomalies in transport propagate meridionally, rather than how water parcels propagate meridionally. "Lagrangian studies have been used to identify when newly formed dense waters from the subpolar gyre reach the subtropics, with anomalies moving with the currents via advection (e.g., Bower et al., 2009; Zou et al., 2016; Jackson et al., 2015). However, transport time series can also adjust more rapidly through a fast boundary-wave mediated response of lower latitude AMOC variability to high latitudes

forcing. Such a response can potentially be identified by lag correlation or coherence analysis of AMOC transport time series, rather than hydrographic anomalies. Based on the increase in subpolar watermass transformation peaking in 2013-2015 and various time lags between the subpolar-to-subtropical AMOC strength determined from numerical simulations, we would anticipate a sign of the increasing subtropical AMOC by 2018-2022."

3) The authors then suggested that a larger AMOC at 45N leads a larger AMOC at 26N by 0-2 years. But using the same Glosea5, Jackson et al. (2016) suggested that the AMOC anomalies at 45N precedes 26N by about 10 years. How to reconcile such a significant discrepancy? Is it related to the use of the observed AMOC at 45N and the modeled AMOC at 26N?

We agree that this is an inconsistency that cannot yet be reconciled from the observations. Given the short duration of the records available and the conflicting time lags identified within GloSea (26N to 45N) itself and between GloSea5 26N and observational estimates at 45N, we are removing the '0-2 year' lag estimate and replacing it with, 'consistent with a possible 0-2 year lag'.

"With the relatively short duration records and the absence of a clear impulse anomaly to track between latitudes, it is not yet possible to identify the timescale of adjustment between the subpolar and subtropical AMOC strength. It appears, however, from comparing the 45°N observational estimate of the AMOC and 26°N from Glosea5, that the adjustment timescale may be short (0-2 years). In contrast, within the GloSea5 reanalysis itself there was a mean lag of 7 years between a peak in Labrador Sea density and the AMOC at 26°N (Jackson et al., 2015). This discrepancy is difficult to reconcile. While GloSea5 has been validated against the 26°N observations, there does not exist an equivalent long AMOC record in the subpolar gyre to verify GloSea5: the OSNAP estimate of the AMOC is too short (21 months) to verify interannual variability of reanalyses (Lozier et al., 2019) and the method used at 45°N with altimetry and gridded hydrography may be subject to errors particular in resolving higher frequency

anomalies at the boundary. "

4) In addition, the authors cited Zou et al. (2019) on the connection between the subpolar UNADW and subtropical LNADW transport anomalies. Note that Zou et al. (2019) suggested that such a latitudinal AMOC connection can be due to gyre-dependent forcing; only strong LNADW transport anomalies can propagate southward from the subpolar region to the subtropics in 4 years. A discussion on this and a reconciliation with the presented analysis are currently missing in the manuscript.

The analysis in Zou et al (2019) relies on a single large anomaly during a relative short model run (1991-2004). They find that for this single large anomaly, that a UNADW transport anomaly in the subpolar gyre leads to a subtropical LNADW anomaly 4 years later. This timescale is, however, inconclusive even within their paper where they have 3 different analysis with three different timescales. We don't believe there is anything substantially new to reconcile with our analysis, as we do not separate 45N into layers, and our AMOC anomalies at 26N are (as they are in Zou et al. 2019, and previous RAPID papers) due to anomalies in LNADW. We have therefore removed this reference to Zou et al. 2019.

5) AMOC-AMV-subpolar OHC relationships. It is a bit confusing about the relationships of AMOC-AMV-subpolar OHC. The authors first suggested that the AMOC lead the AMV byâĹij5 years as shown in a high-resolution model (Moat et al. 2019), but then pointed out the AMOC maximum at 45N precedes the AMV byâĹij10 years in Glosea5 (lines 243-244 in this manuscript). Does it imply that the AMOC-AMV relationship is just model-specific? Moat et al., (2019) shows that in a high resolution model the AMOC leads the AMV by $\sim$5 years at 26N and $\sim$9 years at 50N (Moat et al. 2019, figure 3a), which broadly agrees with the reviewers comment above. Although Moat et al., (2019) found these correlations to be significant at the 95% level, they do not account for all the AMV variability ($R^2 = 0.33$) and other processes could contribute to the variability independent of the AMOC, e.g. Atmospheric teleconnections from the tropics, and variability of the Arctic sea ice and snow cover. From this study the

AMOC leading does seem to be robust. Given the short length of the time series in observations we cannot yet be sure about the absolute lag between the AMOC at 45N and 26N. Here we are presenting the broad scale response of the North Atlantic to changes in the AMOC at 26N.

In addition, how does the AMOC-AMV relationship relate to the subpolar OHC changes? A paper on the full heat budget of the North Atlantic is currently being written by the authors, so we have removed the discussion on the ocean heat content changes from this manuscript.

6) The authors suggested a relationship between the weakened heat transport in the sub-tropics (i.e., in relation to a weak AMOC state) and the cooling subpolar gyre during 2013-2015. Should it be focused on the heat transport at 45N that is at the southern boundary of the subpolar gyre?

45N is spanning the subpolar gyre and intergyre-gyre region, so there is no clear break between the subtropical and subpolar gyres. While 26N is near the middle rather than the north of the subtropical gyre (Fig 1), it is expected that AMOC fluctuations in the subtropical gyre are coherent, so that the heat transport through the middle of the subtropical gyre is proportional to the heat transport through the northern edge of the subtropical gyre (Zhang 2010, Bingham & Hughes 2009).

Another manuscript is in preparation to do a detailed heat budget for the North Atlantic. We are therefore reducing references to the heat transport variability, including removing the OHC time series in Fig 6b.

7) The AMOC at 45N appears to be strengthening after 2011 (Figure 6a), indicating an increasing northward heat transport during the cooling period. How to exclude the impact from the strengthened atmospheric forcing during 2013-2015 (e.g., de Jong and de Steur 2016)? My suggestion is to add a timeseries of the surface heat flux over the subpolar region during the overlapping period of 1985-2018 and discuss accordingly their potential impact on the oceanic changes. Otherwise, in my opinion, it is hard to

draw any conclusions on how the AMOC changes lead the changes in the subpolar OHC.

Desbruyeres et al. (2019) discuss the heat budget in the North Atlantic. They use the time-accumulated MHT relative to a reference period from 1996-2013 and determine that the OHC anomaly is initially entirely explained by MHT, and then (during the development of the cold blob) is not. The time-accumulated quantity is, however, sensitive to the choice of reference period; using a different reference period (1993-2017) results in a change in slope of the time-accumulated quantity (integral of a constant with time is a trend). As we are presently involved in another, more detailed, heat budget analysis, we don't believe we can add significantly to what Debruyeres et al. (2019) already showed.

Other Comments: 8) Lines 162-163: To utilize the lengthy record, the authors could put error bars on the monthly values and comment on how robust the seasonal cycles are.

This has been done. we have updated figure 3 and added the following to the text: "There is a substantial seasonal cycle with an amplitude of 2.0±0.16 Sv and 0.7±0.16 Sv (mean and standard deviation from Monte Carlo estimation) for the annual and semi-annual harmonic, explaining 11% and 2% of the variance, respectively. The residual timeseries, likewise, retains substantial variability with a range of 21.6 Sv and a standard deviation of 3.4 Sv. About 20% of the residual variance is associated with the estimated error of ±1.5 Sv for the 10-day binned data."

9) Lines 180-181: Need more information on Figure 5. How to understand the different change points defined by Mean+CP and Trend+CP? Mean+CP shows an earlier change point around 2008. Also, it is not clear from Figure 5 why Mean+AR(1)+CP is the overall best fit. Please add more details on how this was determined. More details explaining the methodology and how the best model is selected have been added in Section 3.2: "For the models with changepoints, we find the number and locations using the pruned exact linear time algorithm (Killick et al., 2012), which performs an exact search considering all options for any possible number of changepoints and select the optimal number/location balancing the overall fit against the length of each segment. The most appropriate model is selected according to the Akaike Information Criteria (AIC). The AIC differences between each model included in the comparison and the model with the smallest AIC are also computed to assess plausibility of all models. As a rule of thumb, a difference larger than 10 indicates that there is essentially no support for a model given the data and the other models at play (Beaulieu & Killick, 2018). To verify sensitivity to the choice of information criterion, the Bayesian Information Criterion for each model is also computed."

Given that the AIC differences between each model and the one with the smallest AIC are all large (>10), we can conclude that no other model amongst those compared fit the data reasonably well.

10) Line 184: The standard deviation clearly varies with the time scales over which it is derived. I would suggest the authors show the standard error in the mean instead, which seems to be more helpful when determining how distinct the time-mean transports are between two years or any two periods. We have quoted the standard error in the text. The values in Table 1 have been left as the standard deviation as this is more relevant to the AMOC variability, but we have added the de-correlation time scales into the Table 1 caption to enable the standard error to be calculated if required.

11) Line 189: Is section 4.2 just about the relationship to 45N? If so, better to be more specific. Title of section 4.2 has been changed to 'AMOC relationship between 26°N and 45°N'

12) Lines 190-203: Please see my main concern point 1). This has been addressed above in point 1).

13) Line 208: Is the timing of the AMOC increase at 45N (2010-2011) sensitive to the size of the filter? The timing should not be sensitive to the filtering. Desbruyeres

et al. (2019) made a quick test on how their comparison AMOC was influenced by the filtering and concluded that: Lowpass filtered time series presented throughout the paper use a 7-year Hanning window and endpoints are therefore truncated at $\pm$ 3 years. The impact of low-pass filtering AMOC and SFOC time series on the lagged auto-correlations were studied by varying the size of the filtering window (0, 3, 5, 7, 9 and 11 years). While the raw annual time series show small correlations at all lags (R < 0.4), maximum correlations for smoothing windows of 3 years and above were reached at a consistent lag of 5-6 years.

14) Lines 213-214: Is the difference in the variability the same between the AMOC at 45N and 26N both in Glosea5? Like the observations the AMOC- Ekman using GloSea5 at 45N (in density space) does have higher variability than Glosea5 at 26N, but Glosea5 at 45N does have slightly less variability than observation at 45N. The standard deviation of the AMOC- Ekman in GlosSea5 at 45N is 1.02 Sv and at 26N is 0.63 Sv. The standard deviation of the observations is 1.58 Sv (45N) and 0.77 Sv (26N). As we do not make reference to the GloSea5 time series at 45N we have removed the line, "With these two time series, the variability in the GloSea5 estimate of AMOC-Ekman at 26°N is more markedly lower than at 45°N."

15) Line 238: The authors appear to emphasize a 5-year time lead by the AMOC. But I couldn't find any observational evidence even in this analysis for such a time lead. The 5 year time lead is from a coupled climate study by Moat et al. (2019), calculated using fields over a 300 year period. Given the short length of the high quality time series at RAPID 26N (2004 to 2018) is it hard to directly show this lag between AMOC and AMV. In this paper (figure 6) using the GloSea5 reanalysis we show that AMOC leading AMV is robust, but there is a bit of variation in the precise lags.

We have rewritten Section 4.3 to make the description of the lead lag relationship between the AMOC, AMV and NAO clearer.

16) Lines 253-255: Please see my main concern #2. This has been addressed above

in point 2) and 15) and in the text.

---

## Author Comment (AC2) · 15 May 2020

We thank the reviewers for their time in commenting on this paper. We have prepared a detailed response to reviewers #1 and #2.

The GloSea5 time series has been extended to the end of 2018 by Laura Jackson at the UK Met Office, and is now included in this paper (red line Figure 6a). Laura has also contributed to the analysis in the updated paper so we have included her as a co-author.

Anonymous referee #2

17) However, I have some issues with the central scientific focus of the paper and felt that the general thrust of the argument was often misguided

[Figure]

We thank the reviewer for their detailed comments, and also their support of the RAPID 26N observations. We agree that they are the best available observations of the continuously varying AMOC, and have edited the text to improve this emphasis. However, one of this reviewer's major disagreements with the manuscript was that we tried to put the RAPID observations in the wider North Atlantic context and that this should not have been the focus of the manuscript.

We disagree. The value of RAPID lies not only in giving the best possible estimate of AMOC transport variability at an individual latitude (of great value as a benchmark for numerical models, ocean reanalyses and ocean dynamics investigations), we believe that as the RAPID time series lengthens it is enabling us to begin to address key climate questions—the raison d'etre for AMOC studies. These include detailed analyses of local causes of variability and regional impacts of variability, but also how the subtropical AMOC responds to buoyancy (rather than wind) forcing, what influence the AMOC has on decadal and longer variations in the Atlantic, and what the relationship is between the AMOC at different latitudes. It is clear that we are only beginning to have a long enough record to address these questions—and not yet to satisfactorily answer them.

Text has been added in two places: ● Section 2.1, first line: "The 14 years of observations at 26°N represent the most complete and longest records of the directly observed AMOC variability currently available." ● Section 2.1, second paragraph: "The use of boundary moorings which sample at high frequency (hourly) enables high frequency (e.g. tidal and mesoscale) variability to be resolved and not aliased (Kanzow et al., 2009)"

18) I think the GloSea5 data is overused and overly trusted to give a realistic representation of the ocean. The authors attempt to reconcile the results with the 45N time series from Debruyeres et al. (2019), but I think too much respect is paid towards these results which are not of comparable stature

We have added text to clarify that the RAPID observations are the most complete and longest record of AMOC variability, but also that the 45N estimates are the longest available subpolar-area AMOC estimates. While they may be flawed, the covariability between buoyancy forcing and AMOC transport estimates in Desbruyeres et al. (2019) provides some confidence, as does the consistency between the overall findings of Desbruyeres et al. (2019) and the OSNAP programme (Lozier et al., 2019) including that watermass transformation east of Greenland is the major driver of subpolar AMOC transport variability. To provide confidence in GloSea5, Jackson et al., (2019) compared the AMOC at 26N and 50N in a large set of reanalyses and finds agreement in the variability.

Minor Comments 19) Line 27: "Comparing the two latitudes, the AMOC at 26◦N is higher than its previous low" this sentence needs to better distinguish spatial and temporal changes. We have replaced:"We have therefore examined the record of transports at 26°N to see whether the AMOC in the subtropical North Atlantic is now recovering from a previously reported low period commencing in 2009. Comparing the two latitudes, the AMOC at 26°N is higher than its previous low." with "Examining 26N, we find that the AMOC is higher than its previous low, though not yet exceeding its long-term mean."

20) Line 35: Slightly clumsy sentence, repetition of "on" This has been changed to; "It drives a large net northward transport of heat, with one petawatt (1 PW = 1015 W) released to the atmosphere between 26°N and 70°N, impacting the climate in the North Atlantic region (e.g. Srokosz et al., 2012) on surface temperatures, precipitation and sea level (Delworth and Mann, 2000)."

21) Line 71: "Guided by". This language ties directly into point 2. RAPID should lead, not follow. We have updated this to: "Based on the RAPID observations and the recent findings at 45°N, we make preliminary investigations into the meridional coherence of the AMOC transport variability between 26°N and 45°N, and the response at 26°N to the impulse forcing in 2013/15."

22) Line 83: missing "to" This has been corrected.

23) Line 89: The heat and freshwater fluxes are mentioned here but neither shown nor discussed. Perhaps a sentence explaining why? We have added: "Here we focus on the volume transport; updated analyses of the heat and freshwater transports are the subject of a separate study."

24) Line 95: Are the CTD-Os a subset of the CTDs? No, they are in addition as the CTD-Os and only sample every 4 hours. We have clarified this in the text.

25) Line107: "net the" 'Net' has been deleted.

26) Line 109: GloSea5 should be mentioned here. The following has been added in Section 2.3 "We also use data from the GloSea5 global ocean and sea ice reanalysis (Blockley et al 2014, Jackson et al 2016), which uses the NEMO GO5 ocean model with a nominal resolution of 0.25° and with 75 vertical layers (Megann et al 2014). It assimilates in-situ and satellite sea surface temperatures; sub-surface ocean profiles of temperature and salinity; sea ice concentration; and sea level anomalies using the NEMOVAR v13 assimilation scheme (Waters et al, 2015). The experiment is described in more detail in Jackson et al. (2016), with a more in-depth comparison to observations and other ocean reanalyses in Jackson et al (2019)."

27) Line 132: missing "use" This has been corrected.

28) Line 150: Should say "Results" Thank you! This has been corrected.

29) Line 158: why is "anti-correlated" repeated inside the brackets? This has been deleted.

30) Line 158: Clarify: there is no correlation information in the spectral plot. This has been clarified : We replaced: "resulting in a reduction of power at the semi-annual frequency in the AMOC strength relative to the UMO. At periods longer than a year, the AMOC variability is dominated by the UMO transport" with "This anti-correlation is the cause of the reduced power at the semi-annual frequency in the total AMOC relative to

the UMO."

Note that an inference about anti-correlation can be made by comparing spectrum of total AMOC with the spectra of the components. If the total AMOC is less than one of the components then there must be some anticorrelation

31) Line 169: Use the LNADW acronym This has been changed

32) Line 172: "that a reductions" This has been fixed

33) Line 186: This sentence could be improved, just state the maximum and minimum values/times for comparison. This has been rewritten for clarity. "The AMOC transport in the 2017/18 year (17.8 $\pm$ 0.39 Sv) is larger than the recent minimum in 2009/10 (13.5 $\pm$ 0.36 Sv), but this does not represent a return to the high AMOC transport values near the beginning of the observational record (2005/06, 20.9 $\pm$ 0.32 Sv)."

34) Line 210: All this tells us is that GloSea5 is dynamically consistent with itself. It could still be wrong. I assume GloSea5 changes are forced by Lab sea deep convection, which we know many models get wrong, even if it does assimilate observations. This is actually based on observations at 45N and GloSea at 26N. We are using the agreement between GloSea at 26N and RAPID at 26N to provide a view (potentially not correct) of what the longer term variability of the AMOC at 26N may have been, and comparing this against the 45N observations assuming—based on their robust agreement with the surface forced overturning in the subpolar gyre—that they are a reasonable estimate of the AMOC at this latitude. As RAPID 26N is the only array providing the length and quality of AMOC observations, we will necessarily need to look to other products to investigate meridional connectivity—at least until the OSNAP observations provide a longer term, high-quality estimate of subpolar overturning.

35) Line 232: You can, but I think this analysis seems uncoupled from the RAPID results. The reviewer is referring to: "we can look more closely at the period of the observations and the longer records of ocean heat content and SSTs to evaluate whether

the observed variations in the Atlantic, as indexed by the AMV, follow the patterns predicted by the numerical simulations." This is an area of significant interest and debate in AMOC/Atlantic community—is the AMOC responsible for fluctuations in the AMV? While the observed transport records are short relative to multi-decadal variability, some of the underlying processes (how the heat transport relates to OHC change) are within scope and may lead to mechanistic understanding of whether and how the AMOC influences the AMV.

36) Line 256: Be quantitative. What is the minimum fraction of the mean? We apologise but we do not understand what the reviewer is referring to.

37) Line 266: This is the first mention of the 34.5S array (in the conclusions). We have removed the reference to 34.5S

38) Line 272: Insert "within are analysis framework". We have chosen not to add this phrase in order to be more concise.

39) Line 288: Perhaps substitute "understanding" for "knowledge". This has been changed.

40) Figure 1: Red text on green very hard to read for colour blind people This has been updated to bold black text.

---

## Author Response (AR2)

**Pending recovery in the strength of the meridional overturning circulation at 26°N**

Ben. I. Moat[1], David. A. Smeed[1], Eleanor Frajka-Williams[1], Damien G. Desbruyères[2], Claudie Beaulieu[3], William E. Johns[4], Darren Rayner[1], Alejandra Sanchez-Franks[1], Molly O. Baringer[5], Denis Volkov[5,6], Laura C. Jackson[7], Harry L. Bryden[8]

We thank the reviewers for their time in reviewing this paper. We have prepared a detailed response to reviewer #1.

**Anonymous referee #1**

1. Lines 311-314 on Page 8: The authors appear to infer the intensity of "localized deep convection" from the intensity of basin-integrated water mass transformation. However, there have been no observations to support such a reversal around 2005 in deep convection intensity in the subpolar basins. Actually, the deep convection in the Labrador and Irminger basins were intensified during the consecutive winters of 2007/2008 and 2008/2009 (Yashayaev et al. 2016; de Jong et al. 2012).

*We believe this to be a comment about lines 219 to 222 at the start of Section 4.2.*
We apologise for the confusion. By the sentence "These localised deep convection events are part of a wider and longer-term intensification in subpolar water mass transformation that was at a minimum in 2005," we are not arguing that convection and water mass transformation are equivalent. Rather, that localised convection events may occur during watermass transformation. This is a distinction also emphasised by the OSNAP results (Lozier et al., 2019). The evidence for the intensification in subpolar water mass transformation came from the subpolar-wide estimates in Desbruyeres et al. (2019), without invoking convection. Indeed, 2008 was a moderately strong convection year (Yashayaev et al., 2016), though not as intense as 2013-2015.

To add clarity to our discussion we have edited the text:
The 2013/14 and 2014/15 winters saw the return of deep convection in the Labrador Sea in two great impulse events (Yashayaev and Loder, 2016). These localised deep convection events are part of wider and longer-term intensification in subpolar water mass transformation **following the** minimum in 2005 (Desbruyères et al., 2019). **While deep convection is not equivalent to water mass transformation (a distinction emphasised by the OSNAP results, Lozier et al., 2019), it is potential consequence of the continued buoyancy loss in the subpolar gyre. The overall intensification** of the light-to-dense water mass transformation rates **since 2005 has led to an** intensification of the AMOC at the southern exit of the subpolar gyre since 2010, **after a delay of 5-6 years,** as found in a recent observational analysis (Desbruyères et al, 2019).

It further raises a question as to whether there is expected to be a delayed overturning response to the well-documented 2007-2009 events. What are the implications for the expectation of such a response to the 2013-2015 events (lines 324-325, page 8)?

The 2008 convection year was stronger than it had been in the years leading up to 2008 (Vage et al. 2009), but was still deemed a 'moderately strong convection year' by Yashayaev et al. (2016). Given the disconnect between convection and overturning strength now highlighted by the OSNAP array, we are focusing more on the intensity (and timeseries thereof) of watermass transformation rather than convection.

*Lines 333 to 334*
*Additionally, the AMOC at 26N does not yet appear to be responding to the intense buoyancy loss in the subpolar gyre in 2013-2015.*

2. Section 4.3 seems to be defocusing as it discusses the overturning's impact over the time scales that may not be resolved by the RAPID time series. The analysis instead mainly relies on the GloSea5 AMOC at 26N to relate AMV and NAO to overturning changes. Thus, the statement about 'observed AMOC variability' (lines 484-486, page 10) is misleading because of the use of the GloSea5 AMOC series. I would suggest the authors discuss the SST and atmospheric changes during the observational time period and their implications for the longer-term AMV or NAO changes.

*We believe this to be a comment about the last paragraph of Section 4.3.*
*Unfortunately, the 14 years of observational AMOC data from RAPID at 26N is too short to relate it to the long term variability of the SST (AMV) and atmosphere (NAO), which is why we had to use the GloSea5 data. The text has been modified to make clear which data sets the AMOC variability has been obtained from.*

[revised manuscript text omitted]